# Characteristics, Influencing Factors, Predictive Scoring System, and Outcomes of the Patients with Nephrotoxicity Associated with Administration of Intravenous Colistin

**DOI:** 10.3390/antibiotics11010002

**Published:** 2021-12-21

**Authors:** Pornpen Sangthawan, Alan Frederick Geater, Surarit Naorungroj, Piyarat Nikomrat, Ozioma Forstinus Nwabor, Sarunyou Chusri

**Affiliations:** 1Department of Internal Medicine, Faculty of Medicine, Prince of Songkla University, Songkhla 90110, Thailand; psangth@yahoo.com (P.S.); surarit747@gmail.com (S.N.); npiyarat@medicine.psu.ac.th (P.N.); nwaborozed@gmail.com (O.F.N.); 2Epidemiology Unit, Faculty of Medicine, Prince of Songkla University, Songkhla 90110, Thailand; alan.g@psu.ac.th

**Keywords:** colistin, acute kidney injury, incidence, influencing factors, prediction model

## Abstract

Multidrug-resistant Gram-negative infection is a major global public health threat. Currently, colistin is considered the last-resort treatment despite its nephrotoxicity. The purpose of this study was to estimate the incidence, characteristics, and influencing factors and to develop a prediction model for colistin-associated nephrotoxicity. A retrospective study was conducted in the university hospital in the South of Thailand from December 2015 to June 2019. A total of 381 patients (median age (IQR) of 64 (51–62) years) were analyzed. Overall, 282 (74%) had nephrotoxicity according to the Kidney Disease: Improving Global Outcomes (KDIGO) classification. In-hospital, 30-day mortality rates and cost of hospital admission were significantly higher among those with nephrotoxicity. Age > 60 years, comorbidities, serum albumin less than 3.5 g/dL, and concomitant nephrotoxic use were significantly associated with colistin-associated nephrotoxicity with adjusted OR (95% CI) 2.01 (1.23–2.45), 1.85 (1.18–3.6), 1.68 (1.09–2.99), and 1.77 (1.10–2.97), respectively. The prediction model for high-risk colistin-associated nephrotoxicity was identified with good overall performance (specificity of 79.6% (95% CI 70.3–87.1) and positive predictive value of 92.1% (95% CI 88.0–95.1)). In conclusion, the incidence of colistin-associated nephrotoxicity was high and incurred significant morbidity, mortality, and economic burden. Our predictive scoring system is relatively simple and useful for optimizing colistin therapy.

## 1. Introduction

Infection due to multidrug-resistant Gram-negative bacteria (MDR-GNB) is a major global public health threat [1,2] that results in substantial mortality and economic burdens [2,3]. Mortality due to MDR-GNB-related infections is associated with various factors, including the appropriate choice of antimicrobial agents and adverse drug reactions [4,5,6]. Hence, considerations on the choice of commercially available antibiotics are challenging in terms of minimizing drug-associated side effects. Colistin, an old lipo-polypeptide cationic antimicrobial agent, has been widely re-introduced to clinical use for the treatment of infections caused by MDR-GNB [7,8]. Several authorities recommend the use of colistin as monotherapy, as well as in combination with other agents in patients with suspected or documented MDR-GNB infection as a last-resort treatment [7,9,10,11]. Although in vitro studies with colistin have yielded promising results, mortality among patients with MDR-GNB infections on colistin therapy is usually high [12]. The explanations for the unfavorable outcomes of colistin include poor tissue penetration as well as nephrotoxicity of this agent. The occurrence of nephrotoxicity due to colistin use remains a limiting factor [13,14]. Adverse reactions of colistin in clinical settings ranged from 25% to 76% [14,15,16,17,18]. This finding is supported by animal studies in neutropenic mouse, which showed an acute renal failure rate of 32% on day 7 and 52% at the end of treatment (approximately 14 days) [15]. Additionally, the development of nephrotoxicity during therapy with colistin has been associated with even unfavorable outcomes for patients with MDR-GNB infections [4,5]. The purposed mechanism of nephrotoxicity of colistin is similar to its antibacterial effect, which increases tubular epithelial cell membrane permeability, causing ions and water influx leading to cell swelling and lysis [1,19]. The role of oxidative injury and apoptosis in renal tubular cell injury and necrotic damage induced by colistin was previously demonstrated [2,20]. Additionally renal pathological findings among those with colistin-induced nephrotoxicity included focal irregular dilatation of tubules, epithelial and polymorphonuclear cell cast formation and degeneration and regeneration of epithelial cells [1,21]. With the unavoidable nephrotoxicity of colistin, and unclear risk factors influencing nephrotoxicity, strategies to avoid this adverse reaction are still insufficient [22,23]. Furthermore, although polymyxin B, a similar lipo-polypeptide antimicrobial agent, is proven to elicit less nephrotoxicity compared to colistin, this agent is currently not available in Thailand [24]. The use of potential nephroprotective agents against colistin-induced nephrotoxicity has not produced an obviously promising benefit [25,26]. Therefore, with a dearth of options of antimicrobial agents against MDR-GNB infection, colistin is still prescribed mainly for these infections. With the problems unresolved, we explored the characteristics and risk factors and created a predicting scoring system for nephrotoxicity among patients who received intravenous colistin. The aim of this study is to avoid negative clinical outcomes through early detection of nephrotoxicity among patients on intravenous colistin therapy.

## 2. Results

### 2.1. Patient Characteristics

During the study period, 381 patients consisting of 233 (61%) males with a median age (IQR) of 64 (51–62) years received colistin. The patient’s enrollment criteria are demonstrated in Figure 1. The median (25th–75th IQR) BMI was 23 (21–24) kg/m^2^, and the main comorbidity was pulmonary diseases (47%), followed by hypertension (41%) and hematologic/solid organ malignancy (36%). Chronic kidney disease at initiation of colistin administration was found in 23% of patients. There were 152 (40%) admitted with emergency indication, and 132 (35%) initially admitted in the intensive care unit. The most frequent site of infection was the respiratory tract (68%), followed by 54 (14%) with bloodstream infection and 53 (14%) with urinary tract infection. *Acinetobacter baumannii* was the main causative pathogen in 60% of cases. Most of the patients (88%) received the loading dose of colistin and the median (IQR) duration of colistin administration was 11 (6–14) days. Inotropic administration was presented in 24% of patients. Apparently, 98%, 14%, and 12% of patients received concomitant B-lactam antibiotics, vancomycin, and fluoroquinolones, respectively. Table 1 presents detailed patient characteristics.

### 2.2. Characteristics of Patients with Nephrotoxicity

The characteristics of 282 patients who developed acute kidney injury after colistin therapy are described in Table 2. According to KDIGO criteria, 282 (74%) had nephrotoxicity, with 14% of patients classified as acute kidney disease (AKD) without acute kidney injury (AKI), 52% of patients classified as stage 1 AKI, 22% as stage 2 AKI, 6% as Stage 3 AKI, and 2% classified as chronic kidney disease (CKD). The most common abnormalities of electrolyte and acid—base disturbance in this study were hyperkalemia, which occurred in 28% of patients, followed by metabolic acidosis observed in 18% of patients, hypocalcemia indicated in 10%, and hyperphosphatemia in 6%. Systemic complications developed in this study were as follows: volume overload (30%), respiratory failure (9%), and uremic symptoms (7%). Most patients (65%) recovered renal function within 3 months after colistin therapy.

### 2.3. Outcomes and Prognosis of Patients Receiving Intravenous Colistin with Acute Kidney Injury

Table 3 presents the unfavorable outcomes in patients who developed nephrotoxicity after colistin administration, compared to those without nephrotoxicity. In-hospital and 30-day mortality rates among those with nephrotoxicity were significantly higher than those without nephrotoxicity. Non-clinical outcomes, including rates of ICU admission and length of stay in ICU, were not significantly different between those with and without nephrotoxicity. However, the median length of hospital stay after survival in patients with nephrotoxicity was significantly longer than those without nephrotoxicity. Furthermore, the cost of hospital admission in patients with nephrotoxicity was significantly higher than those without nephrotoxicity. These findings reflected a high burden in terms of medical and financial resources among patients who developed nephrotoxicity after colistin administration.

### 2.4. Risk Factors and Prediction Model for Colistin-Associated Nephrotoxicity

Univariable and multivariable risk factors for nephrotoxicity in patients receiving colistin are shown in Table 4. Four variables—age > 60 years, comorbidities, serum albumin less than 3.5 g/dL, and concomitant nephrotoxic use—were significantly associated with nephrotoxicity associated with colistin administration with adjusted OR (95% CI) 2.01 (1.23–2.45), 1.85 (1.18–3.36), 1.68 (1.09–2.99), and 1.77 (1.10–2.97), respectively. From the coefficients of significant predictors in the model, scores were allocated to the predictors and summarized to give a final predictive score. The scores for each diagnostic indicator were age > 60 years = 1, comorbidities = 1, serum albumin < 3.5 g/dL = 1, and concomitant nephrotoxic agent use = 1; otherwise, the allocated score was 0. The scoring system is demonstrated in Table 5. The receiver operating characteristic (ROC) curve of the scoring model shows an area under the curve (AUC) of 0.81 (Figure 2). With an estimated relative net cost of false positive, negative diagnosis, and the prevalence of colistin-associated nephrotoxicity in the target population of 74.3%, a cut-off value of 3 was selected as the optimal point. This cut-point had a specificity of 79.6% (95% CI 70.3–87.1) and a sensitivity of 82% (95% CI 77.0–86.3). With this cut-off value, the positive predictive value and negative predictive values were 92.1% (95% CI 88.0–95.1) and 60.5% (95% CI 51.5–69.0), respectively. The positive likelihood ratio was 4.02 (95% CI 2.71–5.96) and the negative likelihood ratio was 0.23 (95% CI 0.17–0.30).

## 3. Discussion

In our study, the rate of nephrotoxicity after colistin administration was 74%. Most patients (65%) recovered renal function within 3 months after colistin therapy. The in-hospital and 30-day mortality rates among those with nephrotoxicity were significantly higher than those without nephrotoxicity. Significantly unfavorable non-clinical outcomes such as median length of hospital stay after survival and hospital admission cost were observed among those with nephrotoxicity. Factors significantly influencing nephrotoxicity among those receiving colistin included age > 60 years, comorbidities, serum albumin less than 3.5 g/dL, and concomitant nephrotoxic use.

The incidence of colistin-associated nephrotoxicity (74%) in this study was relatively high compared to previous studies. The rates of nephrotoxicity after colistin administration varied with a wide range from incidence previously reported 25–76% [14,15,16,18]. The discrepancies in reported rates among studies are possibly due to differing nephrotoxicity definitions, the severity of illness, and characteristics of patients [14,16,18]. According to the study setting of tertiary care and referral center, it might be explained that the population in our study had a relatively severe illness, with 40% admitted with emergency indication, and 35% with initial ICU admission. The study by Rocco et al. found that the development of nephrotoxicity among patients receiving colistin was strongly correlated with the presence of septic shock and the severity of the patients [27]. Another study that reported a lower incidence of colistin-associated nephrotoxicity enrolled younger patients, with a 45% incidence of nephrotoxicity in 66 patients receiving colistin. The mean age of patients included in the study was 27 (12) years, which was much younger than patients in our study (median age (IQR) of 64 (51–62) years) [28]. Thus Taejaroenkul et al. also reported a high incidence of colistin-associated nephrotoxicity (70.8%) and a median age of 67 years for patients [29].

Pre-existing renal impairment might offer explanations for the different rates of colistin-associated nephrotoxicity [5]. Therefore, a previous study included only patients with normal renal function, and the incidence of nephrotoxicity was relatively low [28]. On the contrary, the percentage of underlying CKD in our study was 23%. Our study is consistent with Omrani et al., who reported 76.1% colistin-associated nephrotoxicity and 23.9% of patients with pre-existing renal impairment [14].

Similarly to previous studies, the development of nephrotoxicity in this study was associated with unfavorable outcomes. Michalopoulos et al. associated the development of nephrotoxicity among patients receiving colistin with significantly high mortality [5]. Our study found that there was significantly higher in-hospital and 30-day mortality among those with nephrotoxicity compared to those without nephrotoxicity. This reflected the poor prognosis of nephrotoxicity associated with colistin. Furthermore, this current study also demonstrated the economic burden of nephrotoxicity associated with colistin administration resulting in a significantly prolonged length of hospital stay after survival and higher hospital costs among those with nephrotoxicity.

Various predisposing factors to nephrotoxicity among the patients receiving colistin have been reported with inconsistent findings. In this study, age > 60 years, comorbidities, serum albumin less than 3.5 g/dL, and concomitant nephrotoxic use were significantly associated with nephrotoxicity. Increasing age was one of the reported significant risk factors for colistin-associated nephrotoxicity. Previous studies have reported consistent findings with our study, indicating increasing age as a significant predictor of colistin-associated nephrotoxicity [16,30,31,32,33,34]. The presence of comorbidities could be a predisposing factor for increased severity of illness, especially among patients with nephrotoxicity. Multiple comorbidities was shown to be associated with the development of nephrotoxicity [35]. Comorbidity by Charlson comorbidity index has been reported as one of the significant risk factors for colistin-associated nephrotoxicity [36] and is consistent with our finding. In this study, the percentage of patients with Charlson comorbidity index > 6 was higher among patients with nephrotoxicity than those without nephrotoxicity and was a significant risk factor for colistin-associated nephrotoxicity after multivariable adjustment (*p*-values 0.004).

Similarly to previous studies, hypoalbuminemia was significantly associated with nephrotoxicity in our study [14,37,38,39]. It has also been demonstrated that severe hypoalbuminemia independently predicted nephrotoxicity during colistin therapy [38]. The exact underlying causal pathway-relating hypoalbuminemia with colistin-associated nephrotoxicity remains unclear. Possible explanations for the increased risk of nephrotoxicity in hypoalbuminemia patients receiving colistin treatment include altering the regulation of fluid distribution, resulting in compromising renal perfusion and colistin ability to clear infection [21,40]. In addition, impairment of renal perfusion and uncontrolled infection can predispose patients to the development of nephrotoxicity. Additionally, hypoalbuminemia might increase the risk of nephrotoxicity by reducing antioxidant activities, lowering the scavenging of reactive oxygen species, and impairing the preservation of renal tubular cells [40,41]. The oxidative and inflammatory pathways seemed to take part in colistin nephrotoxicity [19,20]. Furthermore, hypoalbuminemia might be an indicator of poor clinical status and high susceptibility to severe infection and subsequent nephrotoxicity [21,38].

Concomitant nephrotoxic agents use has been reported as a significant risk factor of nephrotoxicity associated with colistin therapy in many studies [42,43,44]. Concomitant vancomycin was strongly associated with colistin-associated nephrotoxicity [45,46]. Taejaroenkul et al. also found a significantly increased risk of nephrotoxicity in patients receiving colistin and concomitant nephrotoxic agents [29]. The percentage of concomitant nephrotoxic agents’ usage among our patients who developed nephrotoxicity was 62%, which was significantly higher than those who did not develop nephrotoxicity (40%). This could be one of the explanations for the high incidence of nephrotoxicity in our study. This finding should raise awareness and concern for clinicians regarding using combinations of potentially nephrotoxic agents especially antibiotics when prescribing colistin.

There are several previously established risk factors for nephrotoxicity among those receiving intravenous colistin who were not detected in this study [14,16,27,32,43,45]. First, the severity scores of the patients were not different since the study was conducted in a tertiary care and referral center, and most of the enrolled patients had relatively severe cases and lacked of variation in severity score. Second, the ICU admission was not associated with renal injury among those receiving intravenous colistin, because the availability of ICU bed admission was relatively constrained; hence, some of the patients with critical illness were admitted in the general ward. Third, dosage and duration of intravenous colistin were not associated with nephrotoxicity since the clinicians discontinued intravenous colistin earlier than the duration of treatment plan when evidence of renal injury was detected.

Our prediction model which used only four variables—age > 60 years, comorbidities, serum albumin < 3.5 g/dL, and concomitant nephrotoxic agents—performed well with ROC 0.81, high specificity, positive predictive value, and likelihood ratio. The positive predictive value of our model was higher than that reported by Phe et al. (92.1% vs. 87.5%, respectively) [47]. This could be explained by the difference in the prevalence of colistin-associated nephrotoxicity between studies.

Several limitations in this study need to be acknowledged. First, this study was a retrospective study; therefore, there were no data regarding the reasons for clinical judgment. Second, this study did not include a control group, which would have reduced selection bias and confounding. We did not compare the renal outcomes of colistin therapy with the other antibiotics such as tigecycline or sulbactam, which have quite similar efficacy against multi-drug-resistance infection but different potential nephrotoxicity. Third, the characteristics of patients in this study were heterogeneous, which could result in different outcomes when compared to other studies. Fourthly, though the study explained the pathophysiology of nephrotoxicity, data on pathological changes from renal biopsy were lacking. Finally, colistin plasma-level monitoring was absent. The role of trough level in relation to colistin-associated nephrotoxicity was reported [15].

However, a strength of this study is the relatively large number of patients enrolled in the study. The findings of risk factors for nephrotoxic such as concomitant nephrotoxic agents use can be modifiable or avoided. The enrollment included patients with CKD, so the findings can be applied with this group of patients which can be infected with multidrug-resistant Gram-negative infection and cannot be avoided to use colistin. Our prediction model performed well with a high positive predictive value and specificity for identifying patients at a high risk of nephrotoxicity after colistin therapy. This model could be used to help clinicians weigh the risk and benefits when commencing colistin therapy with close monitoring of renal function and consideration of alternative antibiotics if the score indicates a high-risk category. In addition, clinicians should be vigilant for deterioration of renal function during colistin therapy, since this predisposes patients to adverse outcomes and ameliorated renal function of the patients.

## 4. Materials and Methods

### 4.1. Study Population

This retrospective study was conducted between December 2015 and June 2019 in Songklanagarind hospital, the largest university hospital in the South of Thailand. A total of 381 patients, identified through the hospital database, were enrolled in the study. Inclusion criteria were patients aged more than 18 years who received colistin intravenously for at least 72 h. In cases of multiple admissions with colistin administration, only the first admission episode was included in the analysis. Patients who were pregnant, had an end-stage renal disease, or received renal replacement therapy at the initiation of colistin administration, were dead within 3 days of colistin therapy, or had incomplete data were excluded. This study was approved by the institutional review boards of the Faculty of Medicine, Prince of Songkla University. (REC.62-173-14-3).

### 4.2. Data Collection and Variables

The demographic data collected for each patient were age, gender, comorbidities, body weight, and height. Illness severity was assessed at the first date of colistin administration using the National Early Warning Score (NEWS), which was based on a simple aggregate scoring system, indicating the severity of illness and risk of adverse outcomes [48]. In addition, data regarding hypotensive episodes, intensive care unit admission, intubation, and ventilator support were also collected. Concomitant drugs included vasopressor, diuretics, non-steroidal anti-inflammatory drugs (NSAIDs), aminoglycosides, beta-lactams, fluoroquinolones, vancomycin, tigecycline, clindamycin, cotrimoxazole, and amphotericin B. Potential nephrotoxic agents included NSAIDs, vancomycin, aminoglycosides, and amphotericin B. The type of organisms and sites of infection were recorded. Baseline laboratory parameters before and closest to the time of colistin administration such as complete blood count, serum creatinine, and liver function test were included in the analysis. The estimated GFR was calculated using the Chronic Kidney Disease Epidemiology Collaboration equation (CKD-EPI) [49].

### 4.3. Colistin Preparation and Dosing

Colistin used in this study was colistimethate sodium. Colistin dosing of 150 or 300 mg was given as the loading dose in 348 patients. In patients with impaired renal function, colistin dosages were adjusted as recommended [22]. Therefore, the following doses varied from 75–150 mg every 8–24 h. Dosage and duration of colistin administered were recorded.

### 4.4. Definitions of Outcomes 

The primary outcome was the rates and risk factors for colistin-associated nephrotoxicity. The diagnosis and stratification of colistin-associated nephrotoxicity were defined as a rising of serum creatinine from baseline using the Kidney Disease: Improving Global Outcomes (KDIGO) classification with the exclusion of the urinary output criterion [50]. The secondary outcomes were renal outcomes (dialysis-dependent, recovery of renal function) and mortality at hospital discharge, 30 days, and 3 months after the start of colistin administration. Non-clinical outcomes included length of intensive care unit stay among those admitted in the intensive care unit and length of hospital stay. We compared length of hospital stay among patients who survive until the date of discharge from hospital. The costs of hospital admission were also recorded.

### 4.5. Statistical Analysis

Qualitative variables between the patients with nephrotoxicity and without nephrotoxicity were expressed with frequency and percentage; quantitative variables were expressed with median and interquartile range (IQR). Fisher’s exact test, χ^2^ test, and Mann–Whitney test were used for significance testing as appropriate. Crude odds ratio with 95% confidence interval (CI) for variables influencing the emergence of nephrotoxicity was performed with univariate analysis. Variables with significant value < 0.2 in univariate analysis or with clinical concern established by previous studies were processed for multivariate analysis. Stepwise backward regression with the most appropriate Akaike information criterion (AIC) was used to develop a prediction model. Scores for each predictive variable were based on coefficients in the final model. Significance of variables association with nephrotoxicity was defined with a *p* value of adjusted odds ratio < 0.05 by rough approximation with Wald test followed by a likelihood ratio test. A receiver operating characteristic (ROC) curve was used to illustrate the diagnostic ability of the fitted model and scoring. The cut-point for the scoring system was selected with consideration of the prevalence of nephrotoxicity and the relative cost of false negative and false positive. The sensitivity and specificity of the scores were determined based on the constructed ROC curve. With this analysis, we assumed that the net cost of a false negative was 2 times the net costs of a false positive and the prevalence of nephrotoxicity in the target population to be the ratio of nephrotoxicity to all other study participants without nephrotoxicity. All analyses were conducted using the R Language and Environment Version 2.14.1 (Songkhla, Thailand).

## 5. Conclusions

The incidence of colistin-associated nephrotoxicity is high and incurred a significant economic burden to the health care system. The predictive scoring system in this study is relatively simple and useful for clinicians to optimize colistin therapy. The interventions to prevent or minimize colistin-associated nephrotoxicity should be further investigated.

## Figures and Tables

**Figure 1 antibiotics-11-00002-f001:**
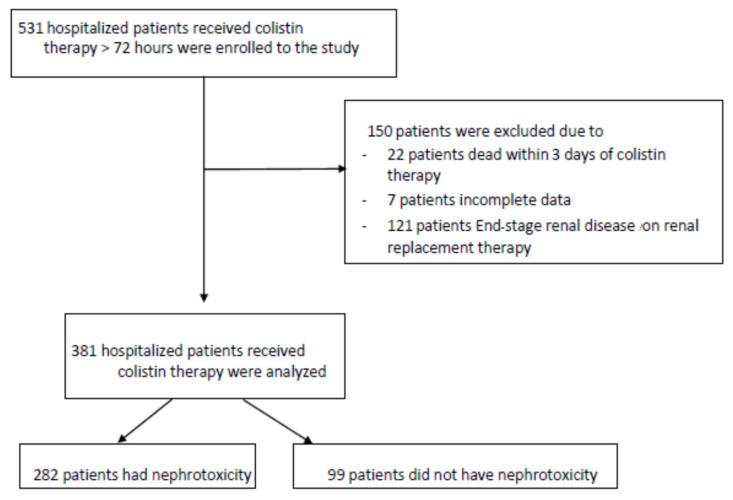
Patients’ enrollment criteria.

**Figure 2 antibiotics-11-00002-f002:**
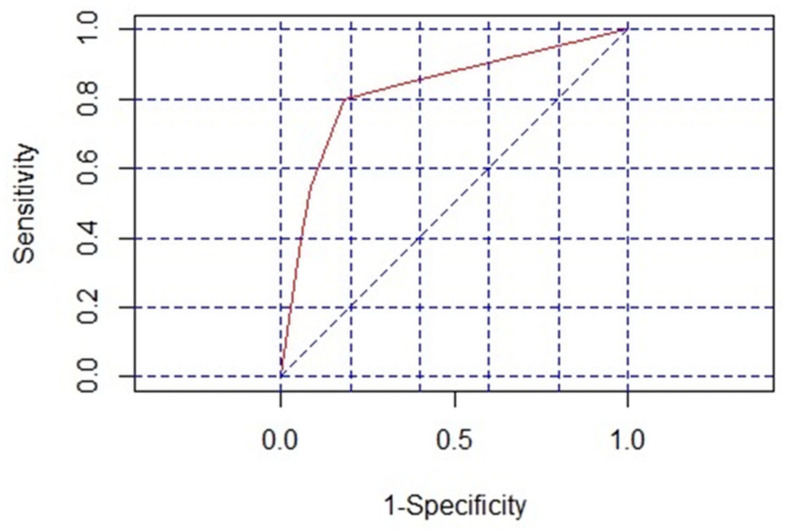
ROC curve (red line) of the scoring model. (blue line: reference line of random chance).

**Table 1 antibiotics-11-00002-t001:** Demographic and baseline characteristics of the patients receiving intravenous colistin.

Parameters	Value(s) for Patients ^a^ (N = 381)
Demographic data	
Age (years), median (IQR ^b^)	64 (51–62)
Male sex	233 (61)
Body mass index (kg/m^2^), median (IQR)	23 (21–24)
Comorbidities	
Coronary artery disease(s)	12 (3)
Congestive heart failure	33 (9)
Hypertension	156 (41)
Diabetes mellitus	79 (21)
Pulmonary disease(s)	178 (47)
Chronic liver disease(s)	70 (18)
Chorionic kidney disease(s)	88 (23)
Cerebrovascular disease(s)	76 (20)
Hematologic/solid organ malignancy	136 (36)
HIV infection	9 (2)
Charlson comorbidity index, median (IQR)	6 (5–8)
Clinical characteristics	
Emergency indication of admission	152 (40)
Initial admission (not mutually exclusive) in intensive care unit	132 (35)
Severity score, median (IQR)	6 (4–9)
Site(s) of infection	
Bloodstream	54 (14)
Respiratory tract	260 (68)
Urinary tract	53 (14)
Musculoskeletal	18 (5)
Gastrointestinal tract	29 (8)
Multiple	30 (8)
Unknown	7 (2)
Causative pathogen(s)	
*Acinetobacter baumannii*	228 (60)
*Pseudomonas aeruginosa*	30 (8)
*Klebsiella pneumoniae*	65(17)
*Escherichia coli*	21 (6)
Concomitant	34 (9)
Unknown	70 (18)
Administration of intravenous colistin	
With loading dosage	335 (88)
Dosage per ABW ^c^ (mg/kg/day), median (IQR)	4.89 (3.78–5.94)
Dosage per IBW ^d^ (mg/kg/day), median (IQR)	4.26 (3.49–5.16)
Cumulative dosage (mg), median (IQR)	2012 (1451–3015)
Duration(days), median (IQR)	11 (6–14)
Concurrent medication	
Inotropic agent/vasopressor	93 (24)
Non-steroidal anti-inflammatory drug	10 (3)
Diuretic agent	148 (39)
Aminoglycoside(s)	10 (3)
Β-lactam antibiotic(s)	376 (98)
Fluoroquinolone(s)	45 (12)
Vancomycin	55 (14)
Tigecycline	6 (2)
Clindamycin	12 (3)
Cotrimoxazole	30 (8)
Amphotericin B	27 (7)

^a^ Values represent number (percent) of patients unless otherwise indicated; ^b^ IQR: interquartile range. ^c^ Actual body weight, ^d^ Ideal body weight.

**Table 2 antibiotics-11-00002-t002:** Characteristics of 282 patients with nephrotoxicity.

Characteristics	Values (N = 282)
**The Kidney Disease: Improving Global Outcomes (KDIGO) classification, *n* (%)**	
Acute kidney disease (AKD) without acute kidney injury (AKI)	40 (14)
Acute kidney disease (AKD) with acute kidney injury (AKI)	
Stage 1	146 (52)
Stage 2	63 (22)
Stage 3	18 (6)
Chronic kidney disease (CKD)	6 (2)
**Electrolyte and acid-base disturbance, *n* (%)**	
Hyperkalemia	78 (28)
Hypocalcemia	29 (10)
Hyperphosphatemia	18 (6)
Acidosis	50 (18)
**Systemic complication, *n* (%)**	
Volume overload	86 (30)
Respiratory failure	24 (9)
Uremic symptoms	19 (7)
**Recovery with eGFR > 60 mL/min/1.73 m^2^ within 3 months, *n* (%)**	182 (65)

**Table 3 antibiotics-11-00002-t003:** Comparisons of outcomes for the patients receiving intravenous colistin with and without nephrotoxicity ^a^.

Outcome	Values for the Patients with Nephrotoxicity (N = 282)	Values for the Patients without Nephrotoxicity (N = 99)	*p* Value ^c^
Mortality, no. (%) of patients			
In-hospital	157 (56)	24 (24)	**<0.001**
14 day	82 (52)	14 (58)	0.735
30 day	152 (54)	78 (79)	**<0.001**
Admission to the intensive care unit	218 (77)	75 (76)	0.861
Length of stay in the intensive care unit	10 (4,20)	10 (5,18)	0.995
Length of hospital stay after survival (days), median (IQR)	59 (45,68)	40 (26,51)	**<0.001**
Cost (baht ^b^), median (IQR)	484,926 (233,266–706,125)	364,956 (213,325–650,683)	**0.048**

^a^ by the Kidney Disease: Improving Global Outcomes (KDIGO) classification; ^b^ 1 USD = 33.16 baht (THB) (as of 14 October 2021); ^c^ Boldface entries indicate values that reached the significance level set at 0.05.

**Table 4 antibiotics-11-00002-t004:** Factors influencing nephrotoxicity among the patients receiving intravenous colistin.

Parameter	Values ^b^		Crude OR (95% CI)	Adjusted OR (95% CI)	*p* Value ^c^
	Patients with Nephrotoxicity ^a^ (N = 282)	Patients without Nephrotoxicity (N = 99)			
Age > 60 years	200 (71)	49 (49)	2.49 (1.55–3.98)	1.92 (1.20–2.21)	**0.031**
Male sex	175 (61)	58 (59)	1.16 (0.73–1.84)		
Body mass index > 25	150 (53)	38 (38)	1.82 (1.14–2.91)	1.12 (0.91–2.01)	0.057
Charlson comorbidity index > 6	102 (36)	15 (15)	3.17 (1.74–5.89)	2.01 (1.34–2.89)	**0.004**
Severity score, median (IQR) ^d^	6 (4,9)	6 (4,8)	1.15 (0.78–2.02)		
Initial admission (not mutually exclusive) Intensive care unit	104 (37)	28 (28)	1.48 (0.90–2.44)	1.16 (0.79–2.01)	0.325
Bloodstream infection	46 (16)	8 (8)	2.23 (1.02–4.92)	1.82 (0.76–3.01)	0.189
Respiratory tract infection	200 (71)	60 (61)	1.59 (0.98–2.56)	1.15 (0.74–1.77)	0.365
Anemia	132 (47)	44 (44)	1.10 (0.69–1.74)		
Elevated transaminase enzyme(s)	107 (38)	39 (39)	0.94 (0.59–1.50)		
Serum albumin < 3.5 g/dL	198 (70)	48 (48)	2.50 (1.57–4.01)	1.59 (1.03–2.71)	**0.042**
Inotropic agent/vasopressor use	70 (25)	23 (23)	1.09 (0.84–1.67)		
Diuretic use	120 (43)	28 (28)	1.88 (1.14–3.09)	1.11 (0.89–2.02)	0.074
Concomitant nephrotoxic agent use	175 (62)	40 (40)	2.41 (1.51–3.85)	1.70 (1.08–2.75)	**0.040**
With loading dosage of colistin	251 (89)	84 (85)	1.45 (0.74–2.81)		
Dosage of colistin per ABW ^e^ (mg/kg/day), median (IQR) ^d^	4.96 (3.88–6.02)	4.81 (3.75–5.89)	1.09 (0.77–1.78)		
Dosage of colistin per IBW ^f^ (mg/kg/day), median (IQR) ^d^	4.42 (3.81–5.29)	4.20 (3.38–5.00)	1.11 (0.91–1.97)	1.03 (0.82–1.81)	0.097
Cumulative dosage of colistin (mg), median (IQR) ^d^	2097 (1502–3124)	1998 (1402–2995)	1.05 (0.74–1.67)		
Duration of colistin administration (day), median (IQR) ^d^	11 (7–15)	10 (6–13)	1.01 (0.58–1.42)		

^a^ by the Kidney Disease: Improving Global Outcomes (KDIGO) classification; ^b^ Values represent number (percent) of patients unless otherwise indicated; ^c^ Boldface entries indicate values that reached the significance level set at 0.05; ^d^ Continuous data; ^e^ Actual body weight; ^f^ Ideal body weight.

**Table 5 antibiotics-11-00002-t005:** Indices of diagnostic scoring system for prediction of nephrotoxicity among the patients receiving intravenous colistin.

Parameter	Diagnostic Index
Age > 60 years	
Yes	1
No	0
Charlson comorbidity index > 6	
Yes	1
No	0
Serum albumin < 3.5 g/dL	
Yes	1
No	0
Concomitant nephrotoxic agent use	
Yes	1
No	0

## Data Availability

All data were included in the manuscript.

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
