# Peer review of "Characteristics, Influencing Factors, Predictive Scoring System, and Outcomes of the Patients with Nephrotoxicity Associated with Administration of Intravenous Colistin"

_antibiotics, 2021, doi:10.3390/antibiotics11010002_

Round 1
Reviewer 1 Report
This paper evaluated the characteristics and influencing factors of colistin-associated AKI and developed its predictive scoring system. The clinical significance of colistin-associated AKI is not small and this study is meaningful, although it is not so novel. Additionally, this study is well designed and has relatively large sample size, some necessary information is missing to interpret the result of this study.
Major concerns and questions for authors:
- Because this paper was regarding the observational study, the authors should describe a STROBE flow chart with the number of patients assessed for eligibility, fulfilling inclusion criteria, excluded, refused to participate (maybe opt out method was applied), lost to follow-up, or consequently analyzed.
- The pathophysiology of colistin nephrotoxicity, the main topic of this paper, should be discussed in more detail with some references in the Introduction and/or Discussion sections.
- In discussing AKI, not all comorbidities are equivalent (eg. pulmonary disease and CKD). As the authors discussed, the use of Charlson comorbidity index is recommended.
- What kind of drugs were included in the “nephrotoxic agent?” Is it only the use of vasopressor, NSAID and diuretic, or does it include other antibiotics? (the methods section said that all antibiotics were included, however, beta-lactams were used in most of the patients and there is a discrepancy between the results in the Table 4)
- How about the relationship between the dose (and/or days) of colistin administration and AKI?
Minor concerns and questions for authors:
- The meaning of “length of hospital stay after survival” was unclear. Does it mean survival after 30 days? Was this analysis performed among the patients who survived after 30 days?
- Wasn’t there any significant relationships between initial ICU admission and colistin-associated AKI? If it was not relevant, then it deviates from the impression in actual clinical practice or evidence. The explanation should be added.
Author Response
Dear Reviewer,
We are very thankful for the valuable comments and suggestions from reviewers and editorial teams. We tried every effort to find out the appropriate response to every comments and suggestions. The responses are following;
Major concerns and questions for authors:
- Because this paper was regarding the observational study, the authors should describe a STROBE flow chart with the number of patients assessed for eligibility, fulfilling inclusion criteria, excluded, refused to participate (maybe opt-out method was applied), lost to follow-up, or consequently analyzed.
Response: We are sorry for the unclear communication on patient enrollment. We have created the STROBE flow chart as recommended. The flow of enrollment of the patients in this study is demonstrated in Figure 1.
- The pathophysiology of colistin nephrotoxicity, the main topic of this paper, should be discussed in more detail with some references in the Introduction and/or Discussion sections.
Response: We have added the sentences as follows; “The purposed mechanism of nephrotoxicity of colistin is similar to its antibacterial effect which increases tubular epithelial cell membrane permeability causing ions and water influx leading to cell swelling and lysis. [1,19] The role of oxidative injury and apoptosis in renal tubular cell injury and necrotic damage induced by colistin was previously demonstrated. [2,20] Additionally renal pathological findings among those with colistin-induced nephrotoxicity included focal irregular dilatation of tubules, epithelial and polymorphonuclear cell cast formation, and degeneration and regeneration of epithelial cells. [1,21] With the unavoidable nephrotoxicity of colistin, and unclear risk factors influencing nephrotoxicity, strategies to avoid this adverse reaction are still insufficient. [22,23]” in the introduction on page 2, Line 50-60.
- In discussing AKI, not all comorbidities are equivalent (eg. pulmonary disease and CKD). As the authors discussed, the use of the Charlson comorbidity index is recommended.
Response: We have added the Charlson comorbidity index in table 1 and have revised the variable of comorbidities to Charlson comorbidity index > 6 in table 2, based on the median value in table 1. Crude and adjusted OR have been changed to 3.17 (1.74-5.89) and 2.01 (1.34-2.89), respectively. We have revised the index of the diagnostic scoring system from comorbidities to Charlson comorbidity index > 6 in table 5, based on adjusted OR in table 2. We have revised the sentences as follows; “[35]. Comorbidity by Charlson comorbidity index has been reported as one of the significant risk factors for colistin-associated nephrotoxicity [36] and is consistent with our finding. In this study, the percentage of patients with Charlson comorbidity index > 6 was higher among patients with AKI than those without AKI and was a significant risk factor for colistin-associated AKI after multivariable adjustment (P-values 0.004). in the discussion on page 8, Line 184-189.
- What kind of drugs were included in the “nephrotoxic agent?” Is it only the use of vasopressor, NSAID, and diuretic, or does it include other antibiotics? (the methods section said that all antibiotics were included, however, beta-lactams were used in most of the patients and there is a discrepancy between the results in Table 4)
Response: We have revised the sentences as follows; “Concomitant drugs included vasopressor, diuretics, non-steroidal anti-inflammatory drugs (NSAIDs), aminoglycosides, beta-lactams, fluoroquinolones, vancomycin, tigecycline, clindamycin, cotrimoxazole, and amphotericin B. Potential nephrotoxic agents included NSAIDs, vancomycin, aminoglycosides and amphotericin B.” in the materials and methods on page 10, Line 261-265. With the understanding that vasopressor and diuretics potentially affect renal function with several mechanisms besides direct nephrotoxicity including volume depletion or decreasing renal perfusion. Then we have analyzed these 2 agents separately from other potentially nephrotoxic agents mentioned above.
- How about the relationship between the dose (and/or days) of colistin administration and AKI?
Response: We have added the data on loading dosage, dosage per actual body weight (ABW), dosage per ideal body weight (IBW), accumulative dosage, and duration of treatment in table 1. We have added these variables into univariate and multivariate analysis in table 4. However, only the variable of dosage of colistin per IBW had P-value in univariate analysis < 0.2 then we have put this variable into the final model. The adjusted OR of this variable was 1.03 (0.82-1.81) with a P-value of 0.097 then we have not included this variable in the diagnostic scoring system for the prediction of acute kidney injury.
Minor concerns and questions for authors:
- The meaning of “length of hospital stay after survival” was unclear. Does it mean survival after 30 days? Was this analysis performed among the patients who survived after 30 days?
Response: We apologize for the unclear communication. We have added the sentence as follows; “We compared the length of hospital stay among the patients who survive until the date of discharge from hospital.” In part of materials and methods on page 11, Line 286-288.
- Wasn’t there any significant relationships between initial ICU admission and colistin-associated AKI? If it was not relevant, then it deviates from the impression in actual clinical practice or evidence. The explanation should be added.
Response: We have added the data on ICU admission into univariate and multivariate analysis in table 4. The crude OR of this variable was 1.48 (0.90-2.44) with P-value in univariate analysis < 0.2 then we have put this variable into the final model. The adjusted OR of this variable was 1.16 (0.79-2.01) with a P-value of 0.325 then we have not included this variable into the diagnostic scoring system for prediction of acute kidney injury. We have added the explanations as follows; “There are several previously established risk factors for nephrotoxicity among those receiving intravenous colistin were not detected in this study. First, the severity scores of the patients were not different since the study was conducted in a tertiary care and referral center, and most of the enrolled patients were relatively severe and lacked variation in severity scores. Second, the ICU admission was not associated with renal injury among those receiving intravenous colistin, because the availability of ICU bed admission was relatively constrained, hence some of the patients with critical illness were admitted in the general ward. Third, dosage and duration of intravenous colistin were not associated with nephrotoxicity since the clinicians dis-continued intravenous colistin earlier than the duration of treatment plan when evidence of renal injury was detected.” in the discussion on page 9, Line 216-226
Reviewer 2 Report
The work is innovative and covers a large number of patients. However, the applied equation must be shown and explained in the methods. How can you apply this model without having a control group? Statistical analysis is not correctly referenced. What statistical programswere used in this work.
Author Response
Dear Reviewer,
We are very thankful for the valuable comments and suggestions from reviewers and editorial teams. We tried every effort to find out the appropriate response to every comments and suggestions. The responses are following;
The work is innovative and covers a large number of patients. However, the applied equation must be shown and explained in the methods. How can you apply this model without having a control group? Statistical analysis is not correctly referenced. What statistical programs were used in this work?
Response: We apologize for the unclear communication. The equation model used in multivariate analysis was explained as follows; “Variables with significant value < 0.2 in univariate analysis or with clinical concern established by previous studies were processed for multivariate analysis. Stepwise backward regression with the most appropriate Akaike information criterion (AIC) was used to develop a prediction model.” in the materials and methods on page 11, Line 310-315.
From table 4, the final equation model using the acute injury as the outcomes and according to the criteria of P values < 0.2, we used Age > 60 years, Body mass index > 25, Charlson comorbidity index > 6, Initial admission (not mutually exclusive) Intensive care unit, Serum albumin < 3.5 gram/dL, diuretic use, Concomitant nephrotoxic agent use and dosage of colistin per IBW as the independent variables to determine factors influencing acute kidney injury among the patients receiving intravenous colistin.
The control group was the patients receiving intravenous colistin without evidence of acute kidney injury as the criteria described in the definition of outcomes in the materials and methods on page 11, Line 290-294.
We have revised the part of statistical analysis as a recommendation as follows; “Qualitative variables between the patients with AKI and without AKI were ex-pressed with frequency and percentage; quantitative variables were expressed with median and interquartile range (IQR). Fisher’s exact test, χ2 test, and Mann–Whitney test were used for significance testing as appropriate. Crude odds ratio with 95% confidence interval (CI) for variables influencing the emergence of AKI was performed with univariate analysis. Variables with a significant value < 0.2 in univariate analysis or with clinical concern established by previous studies were processed for multivariate analysis. Stepwise backward regression with the most appropriate Akaike information criterion (AIC) was used to develop a prediction model. Scores for each predictive variable were based on coefficients in the final model. Significance of variables association with AKI was defined with a P-value of adjusted odds ratio < 0.05 by rough approximation with Wald test followed by a likelihood ratio test.” In the materials and methods on page 11, Line 305-315.
For the diagnostic scoring system for the prediction of acute kidney injury, we have described the process as follows; “A receiver operating characteristic (ROC) curve was used to illustrate the diagnostic ability of the fitted model and scoring. The cut-point for the scoring system was selected with consideration of the prevalence of AKI and the relative cost of false negative and false positive. The sensitivity and specificity of the scores were determined based on the constructed ROC curve. With this analysis, we assumed that the net cost of a false negative was 2 times the net costs of a false positive and the prevalence of AKI in the target population to be the ratio of AKI to all other study participants without AKI.” In the materials and methods on page 11, Line 312-319.
We used R Language and Environment Version 2.14.1 (Songkhla, Thailand) for analysis in this study.
Round 2
Reviewer 1 Report
All of the issues that I pointed out were adequately revised. I have no additional comments.
Author Response
Dear Reviewer,
Thank you very much for the comments. We have tried our best effort to response to the suggestions to improve the manuscript. We have sent the manuscript to our department of international affairs to improve English language and style as well as spell check.
Sarunyou Chusri, M.D., Ph.D.
Corresponding author
Reviewer 2 Report
Changes and comments that were shown were considered adequate and sufficient.Accept in present form.
Author Response
Dear Reviewer,
Thank you very much for the value comments. We have tried every best effort to response the suggestion to improve the manuscript.
Sincerely yours,
Sarunyou Chusri, M.D., Ph.D.
Corresponding author